# Development of Bag-1L as a therapeutic target in androgen receptor-dependent prostate cancer

Laura Cato[1,2], Antje Neeb[3†], Adam Sharp[4], Victor Buzón[5], Scott B Ficarro[6], Linxiao Yang[7], Claudia Muhle-Goll[8,9], Nane C Kuznik[3], Ruth Riisnaes[4], Daniel Nava Rodrigues[4], Olivier Armant[3,10], Victor Gourain[3], Guillaume Adelmant[6], Emmanuel A Ntim[3], Thomas Westerling[1,2], David Dolling[11], Pasquale Rescigno[4], Ines Figueiredo[4], Friedrich Fauser[12], Jennifer Wu[1,2], Jaice T Rottenberg[1,2], Liubov Shatkina[3], Claudia Ester[3], Burkhard Luy[8,9], Holger Puchta[12], Jakob Troppmair[13], Nicole Jung[9], Stefan Bräse[3,9], Uwe Strähle[3], Jarrod A Marto[6], Gerd Ulrich Nienhaus[3,7,14,15], Bissan Al-Lazikani[16], Xavier Salvatella[5,17], Johann S de Bono[4], Andrew CB Cato[3*], Myles Brown[1,2*]

[1]Department of Medical Oncology, Dana-Farber Cancer Institute and Harvard Medical School, Boston, United States; [2]Center for Functional Cancer Epigenetics, Dana-Farber Cancer Institute, Boston, United States; [3]Institute of Toxicology and Genetics, Karlsruhe Institute of Technology, Karlsruhe, Germany; [4]Prostate Cancer Target Therapy Group, Institute of Cancer Research and The Royal Marsden NHS Foundation Trust, Sutton, United Kingdom; [5]Institute for Research in Biomedicine, The Barcelona Institute of Science and Technology, Barcelona, Spain; [6]The Blais Proteomics Center, Dana-Farber Cancer Institute, Boston, United States; [7]Institute of Applied Physics, Karlsruhe Institute of Technology, Karlsruhe, Germany; [8]Institute for Biological Interfaces, Karlsruhe Institute of Technology, Karlsruhe, Germany; [9]Institute of Organic Chemistry, Karlsruhe Institute of Technology, Karlsruhe, Germany; [10]Institut de Radioprotection et de Sûreté Nucléaire, PRP-ENV/SERIS/LECO, Cadarache, France; [11]Clinical Trials and Statistics Unit, Institute of Cancer Research, London, United Kingdom; [12]Botanical Institute II, Karlsruhe Institute of Technology, Karlsruhe, Germany; [13]Daniel-Swarovski Research Laboratory, Department of Visceral, Transplant and Thoracic Surgery, Innsbruck Medical University, Innsbruck, Austria; [14]Institute of Nanotechnology, Karlsruhe Institute of Technology, Karlsruhe, Germany; [15]Department of Physics, University of Illinois at Urbana-Champaign, Urbana, United States; [16]Cancer Research UK Cancer Therapeutics Unit, The Institute of Cancer Research, London, United Kingdom; [17]ICREA, Passeig Lluís Companys, Barcelona, Spain

**\*For correspondence:**
andrew.cato@kit.edu (ACBC);
myles_brown@dfci.harvard.edu (MB)

**Present address:** [†]Prostate Cancer Target Therapy Group, Institute of Cancer Research and The Royal Marsden NHS Foundation Trust, Sutton, United Kingdom

**Competing interests:** The authors declare that no competing interests exist.

**Abstract** Targeting the activation function-1 (AF-1) domain located in the N-terminus of the androgen receptor (AR) is an attractive therapeutic alternative to the current approaches to inhibit AR action in prostate cancer (PCa). Here we show that the AR AF-1 is bound by the cochaperone Bag-1L. Mutations in the AR interaction domain or loss of Bag-1L abrogate AR signaling and reduce PCa growth. Clinically, Bag-1L protein levels increase with progression to castration-resistant PCa (CRPC) and high levels of Bag-1L in primary PCa associate with a reduced clinical benefit from abiraterone when these tumors progress. Intriguingly, residues in Bag-1L important for its interaction with the AR AF-1 are

within a potentially druggable pocket, implicating Bag-1L as a potential therapeutic target in PCa.

DOI: https://doi.org/10.7554/eLife.27159.001

## Introduction

The androgen receptor (AR) plays a central role in the development of prostate cancer (PCa). Current approaches aimed at reducing persistent AR signaling either inhibit the production of androgens, or compete with endogenous ligands for binding to the AR C-terminal ligand-binding domain (LBD) (*Helsen et al., 2014*). While these therapies are initially effective in the advanced disease setting, PCa will eventually progress to a lethal, therapy-resistant state termed castration-resistant PCa (CRPC). In most cases of CRPC, AR continues to play a dominant role. Newer androgen synthesis inhibitors such as abiraterone and AR antagonists such as enzalutamide have been developed as second-generation therapies for the treatment of CRPC (*de Bono et al., 2011*; *Scher et al., 2012*; *Beer et al., 2014*). Nevertheless, while initially effective in the treatment of CRPC, most cases will develop resistance to these therapies (*Mostaghel et al., 2014*). One of the major drawbacks of these drugs is that they target the AR LBD (either directly or indirectly). Although the AR LBD is often regarded as the major site of AR regulation, most of the AR transactivation function is controlled through its N-terminal activation function 1 domain (AF1), which is subdivided into the tau-1 (amino acids 100–359) and tau-5 regions (amino acids 360–528) (*Claessens et al., 2008*). Furthermore, multiple AR variants expressed in CRPC completely lack a LBD (*Guo et al., 2009*), illustrating the need for alternative modes of AR inhibition. Targeting the AR AF-1 therapeutically has been challenging, due to its intrinsically disordered nature and lack of enzymatic activity or rigid binding clefts (*Lavery and McEwan, 2008*). However the lack of secondary or tertiary structure of intrinsically disordered regions (IDRs) of proteins, such as those found in the AR AF-1 domain, could be an advantage in providing a large surface area for protein-protein interactions (*Wright and Dyson, 2009*). IDRs can fold upon binding to their targets, allowing them to undergo conformational changes and participate in protein complex formations (*Dyson and Wright, 2005*). Although these interactions tend to be more transient and of lower affinity than complex formation between structured protein regions (*Latysheva et al., 2015*), they have become exceedingly important for controlling the function of IDR-containing proteins. Proteins that bind the unstructured AR AF-1 domain may constitute regulatory targets for inhibiting AR action.

Bag-1 (Bcl-2-associated athanogene-1) is a multifunctional protein involved in a number of key cellular processes including proliferation, differentiation, cell cycle, transcription and apoptosis (*Townsend et al., 2003a*). Its main function is as a cochaperone and nucleotide exchange factor for Hsp70/Hsc70 (*Alberti et al., 2003*). Four Bag-1 isoforms (Bag-1L, −1M, −1 and −1S) exist in humans and are generated from the same mRNA by a leaky scanning mechanism (*Takayama et al., 1998*; *Yang et al., 1998*). Although the Bag-1 family members differ in their N-terminal domains, their C-terminal (BAG) domains are conserved and essential for interaction with Hsp70/Hsc70 (*Brehmer et al., 2001*; *Sondermann et al., 2001*). Bag-1L, the largest family member, is the only one that possesses a nuclear localization sequence and is therefore localized in the nucleus (*Takayama et al., 1998*) where it enhances the transactivation function of several nuclear hormone receptors, including the AR (*Froesch et al., 1998*; *Knee et al., 2001*; *Shatkina et al., 2003*; *Jehle et al., 2014*). The function of Bag-1L on AR action is mediated through the direct interaction between two regions of each protein. We have recently shown that Bag-1L uses a N-terminal duplicated GARRPR motif to bind to a pocket near the AR LBD, termed binding function-3 (BF-3) (*Jehle et al., 2014*). Additionally, Bag-1L binds via its C-terminal BAG domain to the AR N-terminal domain (NTD) (*Shatkina et al., 2003*). However, details of this interaction and its consequences are unknown.

Here we show that the conserved BAG domain within the C-terminus of Bag-1L selectively interacts with the intrinsically disordered, but partially folded N-terminal AR tau-5 domain. Disrupting this interaction by knocking out or mutating the BAG domain of Bag-1L alters the structural properties of the AR NTD and receptor folding, and reduces the ability of AR to bind to chromatin and regulate transcription.

**eLife digest** Prostate cancer is the second most common cancer in men around the world. The cancer relies on a protein called the androgen receptor in order to develop and grow. Currently, some of the most common treatments for prostate cancer, especially in its advanced stages, are drugs that block the activity of this receptor. However, such treatments are only successful for a limited period of time, and so alternative methods to inhibit this receptor are still needed.

The androgen receptor must bind to a number of proteins to carry out its activity. These proteins include one called Bag-1L, which is also important for the development of prostate cancer. Stopping such a protein from binding with the androgen receptor might represent a new way to treat prostate cancer; but first it will be important to understand how this interaction actually regulates the activity of the receptor.

Now, Cato et al. have analyzed samples of cancer cells that had been collected from 43 patients with prostate cancer and found that Bag-1L levels increase as the disease progresses. Looking at the patients' medical records then revealed that therapies targeting the androgen receptor were less effective in people with high levels of Bag-1L. Conversely, altering, removing or inhibiting Bag-1L in prostate cancer cells grown in the laboratory made the receptor less active and made the cells grow slower.

Further experiments went on to reveal that Bag-1L interacts with a regulatory region of the androgen receptor. Cato et al. note that this region remains largely unexplored therapeutically, because it has some unique structural properties that restrict how much it can interact with drug molecules. Targeting Bag-1L and stopping it from binding to this region of the androgen receptor would represent a different approach to inhibiting the androgen receptor and treating patients with prostate cancer. Together these new findings should provide pharmaceutical companies with much of the information they would require to immediately start screening for therapies that target Bag-1L. Ultimately, Cato et al. hope that any follow-up findings will benefit prostate cancer patients by improving the currently available treatments.

DOI: https://doi.org/10.7554/eLife.27159.002

## Results

### Bag-1L is important for AR activity and function in PCa

Bag-1L has been reported to upregulate the activity of AR (*Froesch et al., 1998*; *Knee et al., 2001*; *Shatkina et al., 2003*). However, previous studies have not been able to clearly separate the function of Bag-1L from that of the other Bag-1 family members (*Krajewska et al., 2006*; *Mäki et al., 2007*). To specifically determine the function of Bag-1L, we employed a transcription activator-like effector nuclease (TALEN) approach that targets the first codon (CTG) of Bag-1L (*Figure 1A*), resulting in the complete knock-out (KO) of this protein. We used this approach in hormone-dependent LNCaP cells where we observed, concomitant with the loss of Bag-1L, an upregulation of the other Bag-1 isoforms (i.e. Bag-1S and Bag-1; *Figure 1B*); this is consistent with the translation of the Bag-1 mRNA by a leaky scanning mechanism. No alterations of AR levels were observed in response to Bag-1L KO, even in the presence of dihydrotestosterone (DHT). We also created rescue cell lines, re-expressing either an empty vector construct or (wild-type) Bag-1L (*Figure 1C*), to exclude potential off-target effects. Cell growth, over a 5 day period, was determined for all cell lines (*Figure 1D*). While loss of Bag-1L significantly reduced PCa growth, re-expression of Bag-1L (but not that of the empty vector) could reverse this phenotype, confirming that LNCaP cell growth is Bag-1L dependent.

Since LNCaP cell growth is primarily driven by AR activity, we next tested whether Bag-1L has any effect on the AR cistrome or transcriptome. We performed ChIP-seq in hormone-depleted control, Bag-1L KO and the rescue LNCaP cell lines, treated for 4 hr with vehicle (ETOH) or 10 nM DHT. We observed only limited binding of AR in response to vehicle treatment, regardless of the presence or absence of Bag-1L (*Figure 1—figure supplement 1*). This is in concordance with previously published AR cistromes (*Wang et al., 2007*). As previously reported (*Wang et al., 2009*), AR binding increased in response to DHT in the control cells (*Figure 1E*). In comparison, increase in AR binding



**Figure 1.** Loss of Bag-1L inhibits PCa cell growth and reduces the AR cistrome and transcriptome. (A) Schematic of the TALEN approach to knockout Bag-1L expression. The TALEN target sequences are highlighted in grey and the start codon (CTG) of Bag-1L is highlighted in yellow. Note, the start codons for the other Bag-1 isoforms remain intact. The resulting coding sequence, which is missing the Bag-1L start codon and flanking regions, is shown in red. (B) Western blot of nuclear extracts from hormone-deprived control (Ctr) and Bag-1L KO (KO) cells treated with vehicle (ETOH) or 10 nM DHT for 4 hr. Protein levels of Bag-1 isoforms (Bag-1L, −1S and −1, but not −1M) and AR are shown. The expression of the nuclear protein Lamin B1 was used for equal protein loading. (C) Western blot of Bag-1L and AR levels in control, Bag-1L KO and Bag-1L KO rescue cell lines, as indicated. β-actin was probed to ensure equal protein loading. (D) Proliferation assay of indicated cell lines grown in complete media and counting on days 0 and 5. Data are the averages of three independent experiments ± SEM, normalized to day 0. p-values were calculated using standard t test; *p≤0.05;

*Figure 1 continued on next page*

*Figure 1 continued*

***p≤0.001; ****p≤0.0001. (E) Venn diagram of AR cistromes from replicate ChIP-seq experiments in hormone-depleted control (red) and Bag-1L KO (blue) cells treated for 4 hr with DHT. The union of binding sites is indicated. (F) Signal profiles and heatmaps of AR ChIP-seq data (centered on the AR peaks) from indicated cell lines. Signals are shown for sites with an AR peak in one or more cell lines. Cells were starved of hormone and treated with 10 nM DHT for 4 hr prior to the experiment. (G) *Top*, fluorescence images of CFP-AR-YFP-expressing hormone-depleted control or Bag-1L KO cells, treated with ETOH or 10 nM DHT for 2 hr. The FRET channel, which is corrected by 17% of $I_{CFP}$, is shown in red, the raw CFP in blue, and the YFP, under direct excitation at 488 nm, in yellow. Results are from two independent experiments with more than 10 images. Scale bar: 10 μm. A schematic representation of the CFP-AR-YFP construct is shown below. *Bottom*, Quantification of CFP-AR-YPF FRET. $I_{FRET}/(I_{FRET} + I_{Donor})$ ratios in the nucleus are shown. Data is the result of readings from more than 25 cells per sample and is presented as mean ±SD. ****p≤0.0001.
DOI: https://doi.org/10.7554/eLife.27159.003
The following source data and figure supplements are available for figure 1:

**Source data 1.** Direct AR-target genes in TALEN control compared with Bag-1L KO cell lines.
DOI: https://doi.org/10.7554/eLife.27159.007
**Figure supplement 1.** Signal profiles and heatmaps of AR ChIP-seq data in Bag-1L control, KO and rescue cell lines in the presence of vehicle.
DOI: https://doi.org/10.7554/eLife.27159.004
**Figure supplement 2.** ChIP qPCR validation of three AR enhancer sites in Bag-1L control and KO cells.
DOI: https://doi.org/10.7554/eLife.27159.005
**Figure supplement 3.** Top five HALLMARK terms (GSEA) associated with direct AR-target genes lost in Bag-1L KO compared to control cells.
DOI: https://doi.org/10.7554/eLife.27159.006

was modest in the Bag-1L KO cells, and peaks were overall weaker than those observed in the control cells (*Figure 1F*). This implies that Bag-1L is necessary for effective AR binding to chromatin. Concomitantly, Bag-1L re-expression, but not the re-expression of the empty vector construct, could restore the AR binding sites (*Figure 1F*). The difference in AR binding in the presence and absence of Bag-1L was independent of the duration of DHT treatment, as confirmed by directed qPCR at three AR-bound enhancers (*Figure 1—figure supplement 2*).

To test if the reduction in AR binding in response to Bag-1L loss has any repercussions on the AR-dependent transcriptome, we next performed RNA-seq in hormone-deprived control and Bag-1L KO cells, treated for 16 hr with ETOH or 10 nM DHT. RNA- and ChIP-seq data were correlated using GenomicRanges (Bioconductor) and direct AR target genes were defined as genes with DHT-induced differential expression (p(FDR)≤0.05, fold change ≥1.5) that harbor a DHT-responsive AR binding site within 50 kb of their transcription start site (TSS). Using this approach, we determined 599 direct AR target genes in the control and 306 in the Bag-1L KO cells (*Figure 1—source data 1*). Using the HALLMARK function from GSEA we established that most of those genes lost in response to Bag-1L KO are associated with 'androgen response' (p=$3.39 \times 10^{-31}$; q = $1.7 \times 10^{-29}$) (*Figure 1—figure supplement 3*), supporting our hypothesis that one of the functions of Bag-1L in PCa is to regulate AR transactivation.

One of the consequences of the loss of a chaperone or cochaperone is an alteration of the folding properties of its client proteins (*Balchin et al., 2016*; *Mayer and Bukau, 2005*). The impaired chromatin binding and transactivation function of the AR in cells lacking the cochaperone Bag-1L may therefore arise from an inability of the AR to adopt the appropriate conformation for its function. To test this, we employed fluorescence resonance energy transfer (FRET) using AR with its N- and C-termini tagged with CFP and YFP, respectively. This allowed us to measure the intra- and inter-molecular AR N/C-terminal interactions associated with the transactivation of the receptor (*Schaufele et al., 2005*). Experiments were carried out in hormone-depleted control and Bag-1L KO cells treated with vehicle (ETOH) or 10 nM DHT for 2 hr, and FRET signals were quantified (*Figure 1G*). We observed a much attenuated (and overall lower) FRET response to DHT in the Bag-1L KO compared to the control cells (i.e., mean difference of 0.24 ± 0.01 in the control versus mean difference of 0.09 ± 0.01 in the KO cells). Taken together, these results suggest that there are significant alterations in the inter-domain interaction and folding of the AR in the absence of Bag-1L that effect the ability of the receptor to efficiently respond to DHT, bind to chromatin and regulate gene expression.

## The BAG domain of Bag-1L binds the tau-5 region of AR

We have previously shown that Bag-1L enhances AR transactivation via direct interaction of the two proteins (*Jehle et al., 2014*). We demonstrated that a novel GARRPR motif, found at the N-terminus

of Bag-1L, interacts with the BF-3 pocket of the AR LBD. We further noted that the C-terminus of Bag-1L binds the AR NTD (*Shatkina et al., 2003*; *Jehle et al., 2014*). To demonstrate that these interactions contribute to AR action, we next performed a series of mammalian one-hybrid assays involving full-length Bag-1L and different domains of AR. Bag-1L enhances the activity of the previously identified AR LBD (in a hormone-dependent manner) as well as the AR AF-1 domain (*Figure 2A*). Moreover, Bag-1L enhances AR activity via the tau-5 domain within the AR AF-1 (*Shatkina et al., 2003*) (*Figure 2B*), a region known for its hormone-independent receptor activity (*Jenster et al., 1995*). We have previously shown that the interaction of AR tau-5 is with the Bag-1L BAG domain, or its flanking regions (*Shatkina et al., 2003*). To specifically delineate which residues, or combinations of residues are involved in this interaction, we employed a SPOT-synthesis technique that allows the screening of a large number of synthetic peptides (*Frank, 2002*). We synthesized short overlapping peptides (21 amino acids each) spanning the entire C-terminal domain of Bag-1L onto a cellulose membrane and incubated it with bacterially-purified GST-tagged AR tau-5. Once specific binding was established by immunoblotting, alanine substitutions were introduced into the synthesis of positively identified spots, until single amino acids were identified in the BAG domain that destroy the interaction with AR tau-5 (*Figure 2—figure supplement 1*). A triple mutation of K231/232/279A, referred to as CMut hereafter, was able to significantly decrease the interaction of AR and Bag-1L in a GST pull-down (*Figure 2C*) and co-IP experiment (*Figure 2D*), and reduce the AR NTD activity in a mammalian one-hybrid assay (*Figure 2E*). Additionally, Bag-1L CMut significantly altered the AR N/C-interaction compared to wild-type Bag-1L (*Figure 2F*), similar to what we observed in the FRET experiments in response to DHT treatment in the Bag-1L KO cells (*Figure 1G*). CMut Bag-1L- compared to wild-type Bag-1L-expressing cells additionally displayed a reduction in AR chromatin binding by ChIP-seq (*Figure 2—figure supplement 2*), similar to that described for the total loss of Bag-1L (*Figure 1F*), and a corresponding reduction in hormone-dependent AR function (*Figure 2—figure supplement 3*). Moreover, we could show that several evolutionarily conserved residues within the BAG domain of Bag-1L (primarily in helices 2 and 3), when mutated, impair the ability of Bag-1L to enhance AR AF-1 transactivation to a similar extent as the CMut Bag-1L protein (*Figure 2—figure supplement 4*). This indicates that mutating the BAG domain disrupts the Bag-1L:AR response similar to what we observed for the complete loss of Bag-1L.

To test if mutations in the BAG domain would disrupt the integrity of associated biochemical complexes, we next employed quantitative, stable isotope labeling with amino acids in cell culture (SILAC) combined with rapid immunoprecipitation mass spectrometry of endogenous proteins (RIME) (*Mohammed et al., 2013*) of LNCaP cells that stably express FLAG-HA-tagged wild-type or CMut Bag-1L. Association of AR, but not Hsp70 (HSPA1), was disrupted in Bag-1L biochemical complexes in the context of the triple mutation (*Figure 2G*); these data agree with the results of our GST pull-down (*Figure 2C*) and co-IP experiments (*Figure 2D*). Several proteins which exhibited decreased association with CMut Bag-1L (*Figure 2—source data 1*) are annotated with functional roles in protein synthesis, localization, or other aspects of proteostasis (*Powers and Balch, 2013*; *Taipale et al., 2014*; *Labbadia and Morimoto, 2015*) (*Figure 2—source data 2*). The dynamics we observed in the biochemical complex as a function of the Bag1L mutant is consistent with our hypothesis that Bag-1L is involved in the folding of AR (*Figures 1G* and *2F*), suggesting a broader role for the BAG domain in proteome homeostasis.

The reduction of interactors for the mutant Bag-1L could, in principle, be the result of an altered BAG domain conformation brought about by the triple mutation. To test this, we recorded $^{13}$C correlation nuclear magnetic resonance (NMR) spectra to compare Cα and Cβ shifts (*Sattler et al., 1999*), which are predominantly influenced by the secondary structure of a protein. Comparison of the Cα and Cβshifts revealed no significant changes in the wild-type and mutant BAG domain peptide (*Figure 2—figure supplement 5*), suggesting that the extent of α-helix formation is essentially unchanged for the two proteins. However, about one third of residues that make the three antiparallel, helix bundles of the wild-type BAG domain (*Briknarová et al., 2001*) shifted to new positions or demonstrated reduced signal intensities in $^{1}$H$^{15}$N-HSQC NMR spectra in response to the K231/232/279A mutations (*Figure 2—figure supplement 6*). This is most likely due to a destabilization of the entire protein caused by the three mutations, a consequence of which is a significant change in the 3D-structure of Bag-1L and hence an altered interactome of CMut compared to wild-type Bag-1L (*Figure 2G*).



**Figure 2.** The Bag-1L:AR interaction is mediated by K231/232/279 in the BAG domain of Bag-1L. (**A, B**) Mammalian one-hybrid assay in HeLa cells transfected with indicated AR domains linked to Gal4 DBD, subjected to increasing concentration of Bag-1L. The results are the mean of three independent experiments ± SEM, relative to the empty Bag-1L expression vector. Schematic representations of the AR domains are shown below. AF-1: Activation function-1; H: Hinge; DBD: DNA-binding domain; LBD: Ligand-binding domain. (**C**) GST pull-down with GST-Bag-1L fusion proteins
*Figure 2 continued on next page*

*Figure 2 continued*

harboring point mutations (as indicated) in their BAG domain and lysates from LNCaP cells. Shown below is a schematic structure of Bag-1L with the triple mutations in the BAG domain, which abolish the interaction with the AR (but have no effect on Hsp70 binding). NLS: Nuclear localization sequence; UBQ: Ubiquitin-like domain; BAG: BAG domain. (**D**) Co-immunoprecipitation of Bag-1L and AR in LNCaP cells stably overexpressing FLAG-, HA-tagged wild-type (WT) or BAG domain mutant Bag-1L (CMut). The IP was performed using an anti-HA-tag antibody against Bag-1L and an antibody against AR and Hsp70 to evaluate binding of these proteins to Bag-1L. Equal protein loading was confirmed by probing for expression of Bag-1L, AR, Hsp70 and β-actin.. (**E**) Mammalian one-hybrid assay in HeLa cells transfected with pG5ΔE4-38 luciferase, TK Renilla luciferase, pM-AR AF-1 and different Bag-1L constructs harboring a wild-type or mutant BAG domain (as indicated). The results are the mean of three independent experiments ± SEM, relative to the empty Bag-1L expression vector. (**F**) Mammalian two-hybrid assay in HeLa cells transfected with Gal4 DBD-AR LBD and VP16-AR-AF-1 and increasing amounts of wild-type (WT) or K231/232/279A mutant Bag-1L (CMut). The results are the mean of three independent experiments ± SEM, relative to the control Renilla luciferase. (**G**) Log-log plot of intensities for proteins detected in forward and reverse SILAC RIME analyses of Bag-1L WT and CMut cells, targeting BAG-1L (dark blue) or IgG (light blue). Black lines represent median IgG-RIME ratios ± 2 standard deviations. Bag-1L, Hsp70 (HSPA1) and AR are indicated in yellow and red, respectively.

DOI: https://doi.org/10.7554/eLife.27159.008

The following source data and figure supplements are available for figure 2:

**Source data 1.** List of Bag-1L interactors altered by the BAG domain mutation (≥2 standard deviations of Bag-1L/IgG control RIME).
DOI: https://doi.org/10.7554/eLife.27159.015
**Source data 2.** Associated functions (FuncAssociate) of Bag-1L interactors altered by the BAG domain mutation.
DOI: https://doi.org/10.7554/eLife.27159.016
**Figure supplement 1.** Schematic of the SPOT synthesis technology.
DOI: https://doi.org/10.7554/eLife.27159.009
**Figure supplement 2.** Overlap between AR cistromes in wild-type and CMut Bag-1L-expressing LNCaP cells.
DOI: https://doi.org/10.7554/eLife.27159.010
**Figure supplement 3.** Top ten GO-terms (GSEA) associated with direct AR-target genes lost in the Bag-1L CMut- compared to the wild-type Bag-1L-expressing cells.
DOI: https://doi.org/10.7554/eLife.27159.011
**Figure supplement 4.** Conserved BAG domain mutations that inhibit the AR AF-1 transactivation.
DOI: https://doi.org/10.7554/eLife.27159.012
**Figure supplement 5.** CBCACONH data of wild-type and CMut Bag-1L.
DOI: https://doi.org/10.7554/eLife.27159.013
**Figure supplement 6.** $^{15}$N-HSQC spectra of wild-type and CMut Bag-1L.
DOI: https://doi.org/10.7554/eLife.27159.014

## The Bag-1L:AR interaction alters the structure of the AR NTD and is druggable

Differences in the structural consequences of the wild-type or mutant BAG domain interaction with the AR AF-1 was next tested using solution NMR spectroscopy. Addition of the wild-type BAG peptide resulted in the reduction of resonance intensities within the C-terminal part of AR AF-1, indicating that these two molecules interact transiently (*Figure 3A*). The residues of AF-1 most affected by this interaction corresponded to tau-5 and were previously identified as partially folded by NMR (*De Mol et al., 2016*), suggesting that the wild-type BAG domain interacts preferentially with this sub-domain. Moderate decreases in intensity were also observed in tau-1, in the region centered around residue 275, which has the propensity to adopt extended conformations. This suggests that although Bag-1L through its BAG domain binds tau-5, the interaction propagates to tau-1. Equivalent experiments carried out with the BAG domain mutant showed a much-attenuated effect, indicating that the strength of the interaction was diminished by the mutations, which agrees with our previous results (*Figure 2*).

Given the importance of the Bag-1L BAG domain for the interaction with AR, as well as its significance for AR function and activity, it is conceivable that inhibition of the Bag-1L:AR interaction through this domain might provide a powerful tool to suppress AR transactivation and PCa growth. We therefore employed the canSAR drug discovery platform (*Bulusu et al., 2014*; *Tym et al., 2016*) to query whether the BAG domain of Bag-1L contains any druggable or ligandable cavities that may be utilized for drug discovery. We analyzed 44 3D structural snapshots of the BAG domain of human Bag-1/Bag-1L (*Figure 3—source data 1*). Although all 44 structures lack a classical 'druggable' site (defined as sites that harbor geometric and physiochemical properties consistent with binding orally-bioavailable small molecules with strict drug-like properties) (*Tym et al., 2016*), we were able to



**Figure 3.** The BAG domain mutations decrease AR binding and overlap a druggable pocket. (A) *Top left*: Normalized peak intensities of AR-AF-1 (residues 142–448) in the presence of GST (grey), or the GST-fused wild-type (blue) or K231/232/279A mutant BAG domain (red) of Bag-1L. A schematic of the τau-1 and τau-5 regions and the nascent secondary structure of these domains are shown. Partially folded helices are indicated by grey cylinders and regions with the propensity to adopt an extended conformation are indicated by black rectangles. *Bottom and right*: Close-up HSQC spectra of

*Figure 3 continued*

representative residues (as indicated) of AR-AF-1 alone (black) and in the presence of GST (grey), or the GST-fused wild-type (blue) or K231/232/279A mutant BAG domain (red) of Bag-1L. (B) 3D-model of the BAG domain of Bag-1L shown in blue, with the predicted druggable pocket highlighted in gold. (C) Rotations of the structure in (B) with the three residues (K231/232/279) within the BAG domain necessary for the interaction with AR indicated in red.

DOI: https://doi.org/10.7554/eLife.27159.017

The following source data is available for figure 3:

**Source data 1.** Distinct 3D-snapshots of the BAG domain of Bag1/Bag-1L.

DOI: https://doi.org/10.7554/eLife.27159.018

identify a large 'ligandable' cavity, which is druggable using peptides or peptidomimetic drugs (*Figure 3B*). Moreover, we could demonstrate that the three amino acids (K231/232/279) important for AR binding lie within (K231 and 232) or just outside the edge (K279) of this cavity (*Figure 3C*). This agrees with our independent findings from the pull-down and luciferase assay (*Figure 2C and E*), where mutation of residues K231 and K232 jointly, but not K279 alone, reduced the binding to AR. However, in these experiments the biggest effect was achieved in response to a simultaneous mutation of all three residues, suggesting that the triple mutant behaves similarly to the complete loss of Bag-1L. The overlap between these experimental results and the computational prediction of the cavity suggests that pharmacological interference with the predicted cavity of Bag-1L is highly likely to impair AR action.

## The Bag-1L:AR interaction can be inhibited by the thioflavin Thio-2

To test if pharmacological interference of the BAG domain of Bag-1L would indeed impair AR action, we tested the efficacy Thio-2, a tool compound BAG domain inhibitor. Thio-2 has previously been postulated to bind to the BAG domain and to inhibit the interaction between Hsp70 and Bag-1 (*Enthammer et al., 2013*; *Papadakis et al., 2016*). A consequence of this is the reduce growth of breast cancer cells, but so far this compound has not been tested in PCa. Thio-2 was indeed able to abrogate the endogenous Bag-1L:AR interaction in LNCaP cells (*Figure 4A*), as well as inhibit the androgen-dependent LNCaP cell growth (IC$_{50}$ = 17.5 µM; *Figure 4B*). Moreover, Thio-2, unlike the AR N-terminal inhibitors EPI-001 or ido-EPI-002 (*Andersen et al., 2010*) and the AR LBD inhibitor enzalutamide, was able to effectively suppress Bag-1L function in a mammalian-1H assay (*Figure 4C and D*). This suggests that targeting Bag-1L through inhibition of its BAG domain, is a therapeutic possibility to abrogate AR function and reduce PCa growth.

## Bag-1L promotes androgen-dependent and -independent PCa growth

To test whether mutation in the BAG domain of Bag-1L (i.e. Bag-1L CMut) could indeed function as a substitute for Bag-1L inhibition or Bag-1L loss, we measured cell growth of Bag-1L KO cells stably expressing an empty vector control, or the wild-type or CMut Bag-1L (*Figure 5A*). Equal protein expression and AR levels were confirmed by western blotting (*Figure 5—figure supplement 1*). While re-expression of wild-type Bag-1L was able to rescue the growth inhibition triggered by Bag-1L KO, re-expression of the vector or Bag-1L CMut failed to do so. Given this result, we postulated that over-expression of wild-type Bag-1L should therefore cause an increase in PCa cell growth. We were able to confirm that overexpression of wild-type, but not CMut Bag-1L, results in an increase in LNCaP growth (*Figure 5B*), without any obvious alterations in AR levels (*Figure 5—figure supplement 2*). Similarly, wild-type Bag-1L overexpression also promotes growth in mouse xenograft experiments, in intact (*Figure 5C*) and castrated animals (*Figure 5D*). However, overexpression of wild-type Bag-1L (*Figure 5—figure supplement 3*), did not cause an increase in CRPC growth in LNCaP-abl and LNCaP95 cells (*Figure 5E and F*), most likely due to endogenously elevated Bag-1L levels in these cells compared to the parental LNCaP line (*Figure 5—figure supplement 4*). Overexpression of the CMut Bag-1L protein on the other hand, lead to a dominant negative effect on cell growth (*Figure 5E and F*), without any obvious alteration in AR level (*Figure 5—figure supplement 5*). This suggests that intact and functional Bag-1L mediates growth of CRPC cell models and may be associated with castration resistance and PCa disease progression.



**Figure 4.** The thioflavin Thio-2 inhibits AR function and AR-dependent PCa growth. (**A**) Co-immunoprecipitation of endogenous Bag-1L and AR in LNCaP cells, treated with or without 5 μM Thio-2 for 16 hr. A Bag-1L-specific antibody was employed for the IP and an antibody against AR was used to evaluate binding. IgG IP was carried out simultaneously as a negative control. One-tenth of the input samples are shown, to confirm equal protein loading. (**B**) LNCaP cells were grown in complete media and treated with indicated concentrations of Thio-2. $IC_{50}$ values were determined after 72 hr using direct cell counts. Data are the averages of three independent experiments. (**C**) Mammalian one-hybrid assay in HeLa cells transfected with pG5ΔE4-38 luciferase, TK Renilla luciferase, pM-AR AF-1 and Bag-1L, and (increasing concentrations of) indicated inhibitors. The results are the mean of four independent experiments ± SEM. p-values were calculated using standard t test; ns: not significant; ***p<0.001. (**D**) MMTV luciferase promoter assay in HeLa cells with MMTV luciferase construct, TK Renilla luciferase and ARΔLBD (amino acids 1–682). The results are the mean of three independent experiments ± SEM. Enza: enzalutamide. p-values were calculated using standard t test; ns: not significant; **p≤0.01.
DOI: https://doi.org/10.7554/eLife.27159.019

## Nuclear Bag-1 is upregulated in CRPC and associates with reduced clinical benefit from abiraterone therapy

Given our hypothesis that Bag-1L is a driver of castration resistance, we next investigated if Bag-1 (and AR) expression levels change with PCa progression. H-scores were determined by immunohistochemistry (IHC) of matched diagnostic (archival) hormone-sensitive PCa (HSPC) and CRPC biopsies (*Figure 6A*) of 43 patients. Bag-1 antibody specificity for IHC was confirmed using Bag-1 siRNA in HeLa cells (*Figure 6—figure supplement 1*), and antibody depletion in various cell lines (*Figure 6—figure supplement 2*) and patient biopsies (*Figure 6—figure supplement 3*). Specificity of the AR antibody for IHC was previously described (*Welti et al., 2016*). Since Bag-1L is the only Bag-1 family member localized to the nucleus, H-scores were determined separately for the nuclear and

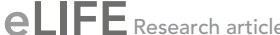

**Figure 5.** Bag-1L enhances hormone-dependent and -independent PCa cell growth. LNCaP Bag-1L KO (Bag-1L KO (**A**)) or parental LNCaP cells (LNCaP (**B**)) were transfected with an empty retroviral construct (+Vector; black), or expression vectors for wild-type Bag-1L (+WT; red) or the Bag-1L K231/232/279A mutant (+CMut; blue). Cells were counted on indicated days by trypan blue exclusion. Data is represented as the mean of three independent experiments ± SEM. p-values were calculated using standard t test; *p≤0.05; **p≤0.01; ****p<0.001. (**C, D**) Cell lines described in (**B**) were injected in 10 intact (**C**) or 9 castrated, athymic nude mice (**D**) and tumor volumes were measured at the indicated time points after injection. Data are represented as mean ±SEM per assay point. p-values were calculated using standard t test; *: p≤0.05; **: p≤0.01. (**E, F**) Cell growth of LNCaP-abl (Abl, (**E**) or LNCaP95 cells (**F**) transfected with an empty retroviral construct (+Vector; black), or expression vectors for wild-type Bag-1L (+WT; red) or the

*Figure 5 continued on next page*

*Figure 5 continued*

Bag-1L K231/232/279A mutant (+CMut; blue). Cells were counted on indicated days by trypan blue exclusion. Data is represented as the mean of three independent experiments ± SEM. p-values were calculated using standard t test; *p≤0.05; **p≤0.01; ****p<0.001.

DOI: https://doi.org/10.7554/eLife.27159.020

The following figure supplements are available for figure 5:

**Figure supplement 1.** Western blot of Bag-1L KO rescue cell lines.

DOI: https://doi.org/10.7554/eLife.27159.021

**Figure supplement 2.** Western blot of Bag-1L over-expression cell lines.

DOI: https://doi.org/10.7554/eLife.27159.022

**Figure supplement 3.** Western blot of wild-type Bag-1L over-expression CRPC lines.

DOI: https://doi.org/10.7554/eLife.27159.023

**Figure supplement 4.** Western blot comparison of endogenous Bag-1L levels in PCa and CRPC lines.

DOI: https://doi.org/10.7554/eLife.27159.024

**Figure supplement 5.** Western blot of BAG domain mutant Bag-1L over-expression CRPC lines.

DOI: https://doi.org/10.7554/eLife.27159.025

cytoplasmic compartments. The Mann-Whitney test was used to compare median H-scores by matched biopsy. While nuclear Bag-1 (i.e. Bag-1L) levels increased significantly (p<0.0001) in the progression from HSPC to CRPC, there was little change in cytoplasmic Bag-1 expression (p=0.14) (*Figure 6B*). In comparison, both nuclear and cytoplasmic AR levels were significantly increased (p<0.0001) as patients progressed to CRPC (*Figure 6C*). Although both nuclear Bag-1 and AR expression increased substantially with PCa progression, expression levels of the two proteins were not correlated (Spearman rank correlation coefficient; *Figure 6—figure supplement 4*).

Given our finding that Bag-1L is a key regulator of AR action (*Figure 1*), we next investigated the correlation between nuclear Bag-1 levels and clinical benefits from AR targeted therapy. Of the 43 patients with matched HSPC and CRPC biopsies, 38 had been treated with abiraterone in the CRPC setting. Of these, 9 (23.7%) were negative and 29 (76.3%) positive for nuclear Bag-1 staining in their primary tumors. There were no significant differences in baseline characteristics of these patient groups at diagnosis or at initiation of abiraterone therapy, except for patient performance status (ECOG PS; p=0.05) (*Table 1*). Moreover, there was no difference in 50% PSA response rate (33.3% vs 29.6%; p=1.00; data not shown), but nuclear Bag-1 positive (compared with Bag-1 negative) patients had a shorter median time to PSA progression (2.8 vs 6.6 months; log rank test p=0.02) (*Figure 6D*) and radiological progression (4.9 vs 9.8 months; log rank test p=0.05) (*Figure 6E*), and a reduced median time on abiraterone treatment (4.9 vs 10.4 months; log rank test p=0.03) (*Figure 6F*). Furthermore, these patients had a decreased median overall survival on abiraterone treatment (15.6 vs 20.9 months), but this was not statistically significant (log rank test p=0.10) (*Figure 6—figure supplement 5*). In contrast, nuclear AR levels, which were positive for all 38 patients at diagnosis, were not associated with time to PSA progression, radiological progression or overall survival (*Table 1—source data 1*). As most of the CRPC biopsies in this cohort were obtained after patients had developed abiraterone resistance (60.5%), neither Bag-1 nor AR expression was predictive of PSA or radiological progression on abiraterone or overall survival (*Table 1—source data 2*). Collectively, these results demonstrate that nuclear Bag-1 (i.e. Bag-1L) increases with PCa progression and is associated with reduced clinical benefit from abiraterone.

## Discussion

Bag-1L is a known regulator of AR action implicated in PCa progression, but its mechanism of action is poorly understood. We show here that Bag-1L is essential for AR transactivation and function, and PCa growth. This function is mediated through the direct interaction of Bag-1L and the AR. Mutations that disrupt this interaction, inhibition of the Bag-1L:AR interaction or loss of Bag-1L altogether, alter the structural properties of the receptor and result in changes in the AR cistrome and transcriptome (*Figure 7*). We show that this leads to reduced growth of hormone-dependent and -independent PCa cells in culture and tumors in xenograft mouse models.

Human Bag-1L is one of four polypeptides translated from a single mRNA by a leaky scanning mechanism (*Takayama et al., 1998*; *Yang et al., 1998*) and as a result it has been difficult to

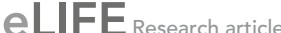

**Figure 6.** Nuclear Bag-1 levels increased from hormone naïve to CRPC status in PCa patients. (**A**) Representative immunohistochemistry images of AR and Bag-1 detection in HSPC and CRPC. Scale bars: 20 µm. (**B, C**) Expression (H-score) of nuclear and cytoplasmic Bag-1 (**B**) and AR (**C**) in 43 matched patient samples at HSPC and CRPC. Median H-score and interquartile range is shown. p-values were calculated using Wilcoxon matched-pair signed rank test. (**D, E**) Kaplan-Meier curves of time to PSA (**D**) or radiological progression (**E**) on abiraterone treatment, for nuclear Bag-1 positive (red; n = 29) and negative (grey; n = 9) patients (at HSPC). Hazard ratios (HR) with 95% confidence intervals (95% CI) and p-values for univariate cox survival model are shown. (**F**) Time on abiraterone for patients, negative (grey) or positive (red) for Bag-1 staining at HSPC. Median time and interquartile range is shown. p-value represents the Mann-Whitney test.

DOI: https://doi.org/10.7554/eLife.27159.026

The following figure supplements are available for figure 6:

**Figure supplement 1.** IHC control in response to Bag-1 knock-down.
DOI: https://doi.org/10.7554/eLife.27159.027
**Figure supplement 2.** Cell line validation of Bag-1 antibody for IHC.
DOI: https://doi.org/10.7554/eLife.27159.028
**Figure supplement 3.** Validation of Bag-1 antibody for IHC using patient samples.
DOI: https://doi.org/10.7554/eLife.27159.029
**Figure supplement 4.** Correlation between nuclear Bag-1 and AR expression in HSPC and CRPC patient samples.
DOI: https://doi.org/10.7554/eLife.27159.030
**Figure supplement 5.** Kaplan-Meier curve of overall survival on abiraterone treatment.

*Figure 6 continued on next page*

*Figure 6 continued*

DOI: https://doi.org/10.7554/eLife.27159.031

demonstrate its function in biological systems, independent of the other Bag-1 proteins. Previous studies on Bag-1L function have either employed siRNA approaches, which reduced the expression of all four isoforms, or Bag-1L overexpression systems (*Froesch et al., 1998*; *Guzey et al., 2000*; *Cutress et al., 2003*; *Shatkina et al., 2003*; *Jehle et al., 2014*). In our present work, we have used genome editing techniques to specifically knock-out endogenous Bag-1L and additionally rescued the knockout by re-expressing wild-type Bag-1L. Using this approach, we could demonstrate that Bag-1L is required for the correct and efficient folding, chromatin binding and transcriptional activity of the AR. Moreover, utilizing mass spectrometry we could show that in addition to binding and regulating the activity of AR, Bag-1L binds stress response proteins, underscoring its function as a survival and antiapoptotic protein and regulator of the proteostasis network (*Townsend et al., 2003b*; *Mosser and Morimoto, 2004*).

We have previously reported that Bag-1L binds the AR through a novel GARRPR motif, found at its N-terminus, and a BF-3 pocket at the AR LBD (*Jehle et al., 2014*). However, inhibition of this interaction by mutating the GARRPR motif had only a modest effect on AR chromatin binding, the AR-mediated transcriptome and PCa cell growth (*Jehle et al., 2014* and unpublished data). Thus, we believe that the Bag-1L GARRPR:AR BF-3 interaction acts as a modulator, rather than a regulator of AR activity. This agrees with findings that the BF-3 pocket itself acts as an allosteric modulator for receptor activity (*Estébanez-Perpiñá et al., 2007*). In addition to the GARRPR:AR BF-3 interaction, Bag-1L also binds the AR N-terminus via its BAG domain. A triple mutant (K231/232/279; CMut) within helices 1 and 2 in the Bag-1L BAG domain was sufficient to inhibit the Bag-1L/AR interaction, in an Hsp70-independent manner. Although other mutations, primarily within helix 3 of the BAG domain, also inhibited the binding of Bag-1L and AR, these sites additionally interacted with Hsp70/Hsc70. The region in AR most affected by binding to Bag-1L maps to the partially folded region within tau-5, which strongly overlaps the region affected by EPI-001 binding (*De Mol et al., 2016*). EPI-001 is a recently developed experimental drug that targets the AR NTD (*Andersen et al., 2010*). A derivative of this compound (EPI-506) is currently being employed in clinical trials for CRPC patients resistant to abiraterone and/or enzalutamide (ClinicalTrails.gov Identifier: NCT02606123). However, EPI-001 and iodo-EPI-002 were not able to suppress Bag-1L function in a mammalian-1H assay. Given our findings that the BAG domain of Bag-1L binds and regulates AR (tau-5) with high specificity, targeting the Bag-1L:AR interaction might be an alternative approach for the treatment of PCa. This notion is supported by our findings pointing to a druggable pocket within the BAG domain of Bag-1L, which overlaps our triple mutation (CMut). An inhibitor (Thio-2) that targets the BAG domain was shown to be efficacious in blocking the antiapoptotic action of Bag-1 in breast cancer and melanoma cells (*Enthammer et al., 2013*; *Papadakis et al., 2016*). In the present study we show that it is also effective in inhibiting the Bag-1L BAG:AR tau-5 interaction and suppressing PCa cell growth.

Aberrant expression of Bag-1 has been described in a variety of human malignancies, such as breast, lung, cervical, colorectal and hepatocellular carcinoma (*Zapata et al., 1998*; *Rorke et al., 2001*; *Clemo et al., 2008*; *Cutress et al., 2003*; *Ni et al., 2013*). Here we show that the nuclear Bag-1 (i.e. Bag-1L), but not the other Bag-1 isoforms, is upregulated in PCa patients that progress from HSPC to CRPC. This is in agreement with previous reports that show that nuclear Bag-1 levels (*Krajewska et al., 2006*), or Bag-1L specifically (*Mäki et al., 2007*), correlates with PCa progression. However, our study is the first to analyze Bag-1 expression in matched HSPC and metastatic CRPC biopsies, rather than unmatched tissues from untreated or hormone-refractory tumors. We additionally show that nuclear Bag-1 levels at HSPC status associate with a reduced clinical benefit from abiraterone. This is in line with previous reports that overexpression of nuclear Bag-1 correlates with drug resistance (*Ni et al., 2013*) and that Bag-1 overexpression is commonly observed in drug-resistant cell lines (*Ding et al., 2000*; *Chen et al., 2002*; *Liu et al., 2009*). Our study provides evidence of an association between Bag-1 levels and treatment response in PCa, highlighting the prognostic significance of Bag-1 in this disease.

**Table 1.** Clinical characteristics of patients at HSPC biopsy and initiation of abiraterone treatment.
HSPC: hormone sensitive biopsy, CRPC: castration-resistant prostate cancer, ECOG PS: Eastern Cooperative Oncology Group performance status, IQR: interquartile range, SD: standard deviation, NA: not available, PSA: prostate specific antigen, n: number, pts: patients.

| | | Overall 38 pts | Bag-1 negative 9 pts | Bag-1 positive 29pts | p-value |
|---|---|---|---|---|---|
| At diagnostic (archival) HSPC biopsy | Biopsy Gleason score, n (%) | | | | |
| | ≤6 | 2 (5) | 0 (0) | 2 (7) | 0.40[†] |
| | 7 | 7 (18) | 4 (44) | 3 (10) | |
| | 8–10 | 27 (71) | 5 (56) | 22 (76) | |
| | NA | 2 (5) | 0 (0) | 2 (7) | |
| | Metastatic at diagnosis, n (%) | | | | |
| | No | 17 (45) | 2 (22) | 15 (52) | 0.25[§] |
| | Yes | 14 (37) | 4 (44) | 10 (34) | |
| | NA | 7 (18) | 3 (33) | 4 (14) | |
| | Primary therapy, n (%) | | | | |
| | Prostatectomy | 4 (11) | 0 (0) | 4 (14) | 0.65[§] |
| | Radiotherapy | 13 (34) | 3 (33) | 10 (34) | |
| | Systemic therapy | 21 (55) | 6 (67) | 15 (52) | |
| | PSA at diagnosis, µg/L | | | | |
| | Median | 46.0 | 61.0 | 29.0 | 0.30* |
| | IQR | 13.1–105.9 | 28.8–150.0 | 10.0–96.6 | |
| At initiation of abiraterone treatment | Age, yr | | | | |
| | Median | 69.2 | 69.5 | 69.0 | 0.59* |
| | IQR | 65.5–73.3 | 61.6–74.1 | 66.1–73.3 | |
| | Sites of metastasis, n (%) | | | | |
| | Node only | 4 (11) | 0 (0) | 4 (14) | 0.57[§] |
| | Bone only | 28 (74) | 7 (78) | 21 (72) | |
| | Visceral (with/without bone) | 6 (16) | 2 (22) | 4 (14) | |
| | ECOG PS, n (%) | | | | |
| | 0 | 12 (32) | 6 (67) | 6 (21) | 0.05[§] |
| | 1 | 24 (63) | 3 (33) | 21 (72) | |
| | 2 | 2 (5) | 0 (0) | 2 (7) | |
| | PSA, µg/L | | | | |
| | Median | 185.5 | 222.0 | 147.0 | 0.77* |
| | IQR | 83.8–445.8 | 51.2–781.5 | 88.0–363.0 | |
| | Hemoglobin, g/L | | | | |
| | Mean | 118.1 | 124.4 | 116.2 | 0.18[‡] |
| | SD | 16.0 | 16.3 | 15.7 | |
| | Alkaline phosphatase, U/L | | | | |
| | Median | 131.0 | 133.0 | 129.0 | 0.85* |
| | IQR | 69.0–230.5 | 64.5–250.5 | 70.0–231.0 | |
| | Lactate dehydrogenase, U/L | | | | |
| | Median | 178.0 | 161.0 | 192.0 | 0.08* |
| | IQR | 155.5–247.0 | 149.5–190.0 | 160.0–287.0 | |
| | NA | 2 | 0 | 2 | |
| | Albumin, g/L | | | | |

*Table 1 continued on next page*

Table 1 continued

| | Overall 38 pts | Bag-1 negative 9 pts | Bag-1 positive 29pts | p-value |
|---|---|---|---|---|
| Mean | 35.8 | 37.8 | 35.2 | 0.17[‡] |
| SD | 4.8 | 2.4 | 5.2 | |
| Previous treatments for CRPC, n (%) | | | | |
| Docetaxel | 27 (71) | 5 (55) | 22 (76) | 0.40[§] |
| Enzalutamide | 2 (5) | 0 (0) | 2 (7) | 1.00[§] |
| Cabazitaxel | 5 (13) | 0 (0) | 5 (17) | 0.31[§] |
| Subsequent treatments for CRPC, n (%) | | | | |
| Docetaxel | 8 (21) | 4 (44) | 4 (14) | 0.07[§] |
| Enzalutamide | 7 (18) | 1 (11) | 6 (21) | 1.00[§] |
| Cabazitaxel | 15 (39) | 5 (55) | 10 (34) | 0.44[§] |

[*]Mann-Whitney test

[†]Chi-square test for trend

[‡]Unpaired t test

[§]Fisher's exact test

DOI: https://doi.org/10.7554/eLife.27159.032

The following source data available for Table 1:

Source data 1. Association of nuclear Bag-1 or AR expression with clinical benefits from abiraterone therapy.

HSPC: hormone sensitive prostate cancer, CRPC: castration-resistant prostate cancer, PSA: prostate specific antigen, HR: hazard ratio, 95% CI: 95% confidence intervals. [a] Univariate cox survival model.

DOI: https://doi.org/10.7554/eLife.27159.033

Source data 2. Clinical characteristics of patients at time of castration-resistant prostate cancer biopsy.

CRPC: castration-resistant prostate cancer, ECOG PS: Eastern Cooperative Oncology Group performance status, IQR: interquartile range, SD: standard deviation, PSA: prostate specific antigen, n: number, pts: patients. [a]t-test from linear regression model of Nuclear Bag-1 H-score at the time of CRPC biopsy [b]Wald test from linear regression model of Nuclear Bag-1 H-score at the time of CRPC biopsy

DOI: https://doi.org/10.7554/eLife.27159.034

In conclusion, we demonstrate here the importance of Bag-1L for AR activity and function in PCa. Combined, our data support targeting the BAG-1L BAG domain:AR tau-5 interaction therapeutically in the treatment of PCa and CRPC.

## Materials and methods

### Cell line preparation and maintenance

TALEN Bag-1L KO and vector controls were created as described (*Cermak et al., 2011*). In brief, the left Bag-1 TALEN (targeting the sequence 'GGGCGGTCAACAAGT') was translated into the RVD code 'NN NN NN HD NN NN NG HD NI NI HD NI NI NN NG' and assembled in pZHY500. The right Bag-1 TALEN arm (targeting the sequence 'CGGGGGGGGCGCGGAGA'), was translated into the RVD code 'HD NG HD HD NN HD NN HD HD HD HD HD HD HD NN' and assembled in pZHY501. Subsequently, both arms were subcloned into pZHY013 (a kind gift from Daniel Voytas) to generate a heterodimeric Fok1 nuclease. The plasmids were then subcloned into pDest12.2 using the Gateway cloning technology (Thermo Fisher Scientific, Waltham, MA) and transiently transfected into LNCaP cells using Fugene 6 (Promega, Madison, WI). Transfected cells were selected using G418 for 24 hr (800 µg/ml). Single clones where isolated by serial dilution and screened for Bag-1L deletion by western blotting. To verify the genomic deletion, part of exon 1 was PCR-amplified and cloned into pcDNA3.1 V5 His6 (Thermo Fisher Scientific) using the TOPO cloning kit (Thermo Fisher Scientific), and sequenced using specific primers (Bag1g14fw 5'-GCTGGGAAGTAGTCGGGC-3'; Bag1g252rev 5'-CTGGTGGGTCGGTCATGC-3'). Stable TALEN rescue cell lines (Bag-1L KO +Vector control, Bag-1L KO +Bag-1L WT and Bag-1L KO +Bag-1L CMut) and stable wild-type (Flag-HA-tagged) Bag-1L overexpressing LNCaP cells were created as previously described (*Jehle et al., 2014*), using expression plasmid poZN. HeLa cell lines were employed for transient transfection



**Figure 7.** The Bag-1L is a promising target for the inhibition of the AR NTD. The function of Bag-1L in PCa is to promote the correct folding of the AR, augment its affinity for chromatin and regulate its transcriptional activity. Loss of Bag-1L, or mutation or inhibition of its BAG domain, leads to the abrogation of these processes and hence reduction of PCa growth.

DOI: https://doi.org/10.7554/eLife.27159.035

using PromoFectin (PromoCell, Heidelberg, Germany) according to the manufacturer's instructions. LNCaP-abl and LNCaP95 cells were a kind gift from Zoran Culig (Innsbruck Medical University, Austria) (*Culig et al., 1999*) and Stephan Plymate (University of Washington, Seattle, WA) (*Hu et al., 2012*), respectively. Unless otherwise stated, all cell lines were obtained from the American Type Culture Collection and their identities were confirmed by short tandem repeat profiling (BioSynthesis, Lewisville, TX). They were all confirmed to be mycoplasma negative, using the MycoAlert mycoplasma detection kit (Lonza, Portsmouth, NH). All cell lines and parental LNCaP cells were cultured in RPMI 1640 medium or DMEM (for HeLa cells only) supplemented with 10% FBS, penicillin (100 u/ml), streptomycin (100 u/ml) and L-glutamine (2 mM). For experiments requiring hormone starvation, cells were grown for 72 hr in phenol red-free RPMI 1640 medium, supplemented with 10% charcoal-stripped FBS, penicillin (100 u/ml), streptomycin (100 u/ml) and L-glutamine (2 mM). CRPC lines were continuously cultured under hormone starvation condition as described above. Cell proliferation experiments were carried out as previously described (*Groner et al., 2016*).

## Animal experiments

All animal experiments were performed per European and German statutory regulations. Animal protocols were approved by the 'Regierungspräsidium' Karlsruhe, Germany (AZ 35–9185.81/G-43/14 'Bag-1L-Prostastakarzinomprojekt'). LNCaP xenograft tumor studies were carried out as previously described for intact (*Maddalo et al., 2012*) or castrated mice (*Eder et al., 2013*).

## Patient cohort, human tumor samples and tissue analysis

Patients were identified from a population of men with metastatic CRPC treated at the Royal Marsden NHS Foundation Trust. All study participants had given written, informed consent and were enrolled in institutional protocols approved by a multicenter research ethics committee (Ethics Committee Centre: London-Chelsea Research Ethics Committee, Reference no. 04/Q0801/60). Forty-three patients with sufficient formalin-fixed, paraffin embedded (FFPE), matched diagnostic (archival) HSPC and CRPC tissue were included in our study. HSPC tissue demonstrated adenocarcinoma and was obtained from either prostate needle biopsy (35), transurethral resection of the prostate (TURP; 3), prostatectomy (4) or bone biopsy (1). CRPC tissue was obtained from the same patients through biopsies of bone (25), lymph node (10), liver (5), prostate (TURP; 1), bladder (1) or chest wall (1). All tissue blocks were freshly sectioned and only considered for IHC analyses if adequate material was present (≥50 tumor cells). For Bag-1 IHC (using antibody Y166, Abcam), HSPC and CRPC FFPE biopsies were first deparafinised, followed by antigen retrieval (microwaving in citrate buffer (pH 6.0) for 18 min at 800 W). The Bag-1 antibody was diluted (1:250) in Dako REAL diluent (Agilent Technologies, Santa Clara, CA) and tissue was incubated for 1 hr. After washes, the bound antibody was visualized using the Dako REAL EnVision Detection System (Agilent Technologies). Sections were counterstained with hematoxylin. AR protein expression was determined using the AR mouse monoclonal antibody (AR441, Agilent Technologies), as previously described (*Welti et al., 2016*). Nuclear and cytoplasmic Bag-1 and AR expression was determined for each case by author D.N.R in a blinded fashion using the modified H-score (HS) method using formula: [(% of weak staining) x 1] + [(% of moderate staining) x 2] + [(% of strong staining x 3) to provide a range from 0 to 300 (*Detre et al., 1995*). HS data was reported as median values with interquartile range (IQR). Demographic and clinical data for each patient were collected retrospectively from the hospital electronic patient record system. These characteristics were compared by Bag-1 status at HSPC using Fisher's exact test for categorical characteristics, the Chi-squared test for trend for ordinal characteristics and either an unpaired t-test for continuous data, if normally distributed (e.g. hemoglobin and albumin), or a Mann-Whitney test. The characteristics were then compared by Bag-1 HS value at CRPC biopsy using linear regression models and either a t-test or Walt test. PSA progression was defined as an increase in the PSA level of 25% or more above the nadir (and by ≥2 ng/ml); patients who stopped abiraterone without PSA progression were censored. Radiological progression was defined as any radiological imaging reporting disease progression; patients who stopped abiraterone without radiological progression were censored. Overall survival was defined as time from initiation of abiraterone to date of death (35 patients) or last follow up/contact (3 patients).

## Protein assays

Co-immunoprecipitation was carried out as described previously (*Jehle et al., 2014*). Proteins were isolated using TIVE lysis buffer (50 mM Tris-HCl pH 7.8, 2 mM EDTA, 150 mM NaCl, 1% NP-40, protease inhibitors), or lysis buffer A (10 mM HEPES pH 7.5, 10 mM KCl, 0.1 mM EGTA, 0.1 mM EDTA, 1 mM DTT, protease inhibitors) plus 0.5% NP-40 and lysis buffer C (20 mM HEPES pH 7.5, 420 mM NaCl, 1.5 mM MgCl$_2$, 0.2 mM EDTA, 25% Glycerol, 1 mM DTT, protease inhibitors) for total, cytoplasmic and nuclear proteins respectively. Western blotting was carried out using standard protocols with following antibodies: Bag-1 (FL-274 and F-7, Santa Cruz Biotechnology, Dallas, TX or Y166, Abcam, Cambridge, UK), Bag-1L (was obtained from Andrew Cato; *Crocoll et al., 2000*), AR (N-20, Santa-Cruz Biotechnology), Hsp70 (K-20, Santa Cruz Biotechnology), Hsp70/Hsc70 (W27, Santa Cruz Biotechnology), Flag-tag (M2, Sigma Aldrich, St. Louis, MO), HA-tag (F-7, Santa Cruz Biotechnology or Abcam), β-actin (C4, Santa Cruz Biotechnology or Abcam), vinculin (V9131, Sigma Aldrich), Lamin B1 (EPR8985, Abcam). Bag-1L and AR inhibitors were produced by Jakob Troppmair (Thio-2), Xavier Salvatella (EPI-001 and iodo-EPI-002) or purchased from Selleckchem (enzalutamide).

## Chromatin immunoprecipitation (ChIP) and ChIP-seq

ChIP was carried out as described (*Jehle et al., 2014*), using anti-AR antibody (N-20; Santa-Cruz Biotechnology). Following primers were utilized for directed ChIP qPCR. KLK3 (AREIII) primers were previously published (*Jehle et al., 2014*).

*TMPRSS2 Fwd:* 5'-GCTCACACAGGATCAGAGCA-3'
*TMPRSS2 Rev:* 5'-TGCTCGTTAGTGGCACATTC-3'

*NKX3.1 Fwd:* 5'-TTTGGGCCACCCTGTAAATA-3'
*NKX3.1 Rev:* 5'-GGGTGGGAGGAGATGAAAAT-3'

ChIP-seq libraries were generated using the ThruPLEX DNA-seq kit (Rubicon Genomics, Ann Arbor, MI) and were sequenced on the Illumina NextSeq 500 platform at the Molecular Biology Core Facility (Dana-Farber Cancer Institute). ChIP-seq data was processed using ChiLin2 (*Qin et al., 2016*). All ChIP-seq data have been deposited at the GEO depository under accession number GSE89939.

## RNA-sequencing

Total RNA was extracted from cells using innuPREP RNA Mini (Analytic Jena AG, Jena, Germany), following the manufacturer's instruction. mRNA libraries were generated using the Illumina TruSeq stranded mRNA sample kit and 1 µg of total RNA per sample. Library preparation, sequencing on a HiSeq1500 Illumina platform, and data analysis were carried out at the NGS facility of the Institute of Toxicology and Genetics (KIT). Fastq files were processed with CASAVA and mapped against the human reference genome GRCh37 using TopHat 2.0.11 (*Trapnell et al., 2009*). Reads were quantified with HTSeq (*Anders et al., 2015*), using the reference gene annotation from Ensembl. Differential expression analysis was performed using DESeq2 (*Love et al., 2014*). All RNA-seq data have been deposited at the GEO depository under accession number GSE89939.

## FRET

Cells were transiently transfected using Lipofectin (Thermo Fisher Scientific) with plasmid CFP-AR-YFP (*Schaufele et al., 2005*), generously provided by Marc Diamond. Images were acquired on an Andor Revolution XD spinning disk laser scanning microscopy system (BFi OPTiLAS) using two color channels. CFP was excited at 405 nm (90 µW, 100 ms) and its emission collected through a 447/60 nm bandpass filter (center wavelength/width, AHF). YFP was excited at 488 nm (187 µW, 100 ms) and observed through a 560/55 nm bandpass filter. A longpass dichroic mirror (DCLP 530, AHF) was used to separate the emission light. In addition, a notch filter (532/10 nm, center wavelength/width, AHF) was inserted in front of the dichroic mirror to reduce crosstalk between CFP and YFP. Additional control experiments were performed to minimize crosstalk and direct excitation of YFP. After background subtraction, FRET signals were calculated using $I_{corr.\ FRET} = I_{FRET} - 0.17\ I_{CFP}$. Nuclear regions were identified manually based on YFP staining. Acquired images were analyzed using ImageJ (*Abramoff et al., 2004*).

## SPOT assay

Short overlapping peptides (21 amino acids each) spanning the entire human Bag-1L BAG domain and its flanking regions were synthesized and spotted in duplicates onto amino-PEG cellulose membranes (Intavis AG Bioanalytical Instruments, Cologne, Germany) using an automated SPOT synthesizer (MultiPep, Intavis AG Bioanalytical Instruments) (*Frank, 2002*). The membranes were incubated with bacterially-purified GST-AR-τau-5 (amino acids 360–528). Specific binding was detected using an anti-GST antibody (Santa Cruz Biotechnologies). Alanine substitutions were introduced into the synthesis of positively identified peptides using site-directed mutagenesis (Agilent Technologies) and the hybridization procedure was repeated until single amino acids were identified in the BAG domain that destroyed the interaction with AR τau-5.

## GST pull-down, mammalian one- and two-hybrid assays

GST pull-down was performed as previously described (*Jehle et al., 2014*). Mammalian one-hybrid experiments were performed as described (*Shatkina et al., 2003*), using constructs pG5ΔE4-38 luciferase (Gal4 reporter gene), TK Renilla luciferase, Gal4 fusion genes (pM-AR-AF-1, pM-τ1AR, pM-τ 5AR) or MMTV luciferase, TK Renilla luciferase and ARΔLBD (amino acids 1–682) with pcDNA3 Bag-1L (wild-type and mutants). Mammalian two-hybrid assays were performed using constructs pG5ΔE4-38 luciferase and TK Renilla luciferase (*Jehle et al., 2014*), and Gal4DBD-ARLBD and VP16-AR-AF-1 (provided by Karin Knudsen).

## RIME

SILAC-labeled rapid IP-mass spectrometry of endogenous protein (RIME) was carried out essentially as previously described (*Mohammed et al., 2013*). Bag-1L WT and CMut cells were grown in media supplemented with 'light' (L-lysine-2 HCl, L-arginine-HCl) or 'heavy' isotope labels ($^{13}C_6$L-lysine HCl, $^{13}C_6^{15}N_4$L-arginine-HCl) and mixed at a 1:1 ratio (30 million cells per treatment arm) prior to IP. Proteins were immunoprecipitated using 20 µg anti HA-tag (Abcam) or rabbit IgG (Santa Cruz Biotechnology) antibody-coupled Dynabeads (Thermo Fisher Scientific). Immunoprecipitated proteins were digested directly on beads as described (*Mohammed et al., 2013*). Recovered peptides were acidified with 10% TFA and subjected to batch mode RP-SCX to desalt peptides and remove traces of detergent (*Adelmant et al., 2011*). After trapping on a self-packed pre-column (100 um I.D. packed with 4 cm POROS 10R2; Applied Biosystems, Foster City, CA), peptides were eluted with an HPLC gradient (0–35% B in 4 hr) and resolved using a self-packed analytical column with integrated ESI emitter tip (*Ficarro et al., 2009*) (30 um I.D. packed with 50 cm Monitor C18, Orochem, with ~1 um ESI tip, flow rate ~30 nL/min) prior to electrospray (voltage = 3.8 kV). Peptides were analyzed by nanoflow LC-MS/MS using a NanoAcquity UPLC system (Waters Corporation, Milford, MA) coupled to an Orbitrap Fusion mass spectrometer (Thermo Fisher Scientific), as previously described (*Ficarro et al., 2009*). The mass spectrometer was operated in data-dependent mode such that the top 15 precursor ions in each MS scan (image current detection, 120K resolution, m/z 300–2000, target 5e5, max inject time = 500 ms) were subjected to both CID (quadrupole isolation, width 1.6 Da, first mass = 110, CE = 30%, rapid scan, target = 5e3, max fill time = 50 ms, electron multiplier detection) and HCD (quadrupole isolation, width 1.6 Da, first mass = 110, CE = 30%, target = 5e4, max fill time = 50 ms, image current detection with 15K resolution). Mascot was used to search peak lists (.mgf) generated by multiplierz (*Askenazi et al., 2009*; *Parikh et al., 2009*) against a forward-reverse database of human proteins (NCBI refseq). Search parameters specified SILAC quantitation (K + 8, R + 10), variable methionine oxidation, and fixed carbamidomethylation of cysteine residues, as well as a precursor mass tolerance of 10 ppm and product ion mass tolerances of 0.6 Da and 25 mmu for CID and HCD spectra. After filtering results to 1% FDR, SILAC quantitation was performed using multiplierz scripts for genes or gene groups represented by at least 2 unique peptides. SILAC data were then normalized according to Bag-1L expression levels (CMut vs. WT). For protein products detected in 'forward' (WT light, CMut heavy) and 'reverse' (WT heavy, CMut light) SILAC experiments, ratios were averaged to provide an aggregate CMut/WT ratio. Search results were each filtered to 1% FDR, combined and further filtered to yield proteins represented by 2 or more unique peptides. Proteins detected in the Bag-1L RIME experiment were considered background and excluded from further analysis unless total summed peptide signal intensities were ≥3 fold higher than those in the control experiment. Genes or gene groups were considered regulated by the mutation of Bag-1L if ratios deviated by more than 2 standard deviations from median-normalized IgG control ratios. Data were plotted using R version 3.2.4.

## NMR

The $^{15}N$-labeled thrombin protease-cleavable GST-fused BAG domain (wild-type or K231/232/279A mutant) was cloned into pGEX-6P vector (Addgene, Cambridge, MA) and expressed in *E. coli* strain BL21 (DE3), grown in M9 minimal medium supplemented with 0.5 g/l $^{15}NH_4Cl$. The $^{15}N$-BAG domains were cleaved off by rhinovirus 3C protease (PreScission, GE Healthcare, Marlborough, MA). To remove residual GST, the protein solution was purified using a glutathione sepharose column and subjected to size exclusion chromatography (Superdex 200, HiLoad 16/60, GE Healthcare). $^{15}N$-HSQC spectra for wild-type and mutant BAG domains (at approximate 500 µM) and standard triple resonance backbone experiments (HNCA, HNCACB, CBCA(CO)NH) for peak assignment were acquired at 23° C on a Bruker Avance I 600 spectrometer. The spectrometer was equipped with a broadband triple resonance probe head with 4 scans per increment and a total of 128 increments in the indirect dimension. For chemical shift calibration and to compare relative signal intensities, 0.2 mM DSS (2,2 Dimethyl-2-silpentane-5-sulfonic acid) was added. Data were processed with NMRPipe (*Delaglio et al., 1995*) and analyzed using NMR VIEW (*Johnson, 2004*).

Expression, purification and nuclear magnetic resonance assignment of isotopically labeled $^{15}N$-AF-1 has been previously described (*De Mol et al., 2016*). [$^1H$,$^{15}N$]-HSQC spectra for AR-AF-1 alone (at a 25 µM concentration), at a 1:5 molar ratio with GST (as a control) or with unlabeled GST-BAG

(wild-type or mutant) were acquired at 278K on a Bruker 800 MHz spectrometer. Data was processed using NMRPipe and NMRDraw, and analyzed using CcpNmr Analysis (*Vranken et al., 2005*). Peak intensities were normalized and plotted as a function of residue number.

### Computational modeling

The 3D structure of human Bag-1L was analyzed using the structure-based druggability algorithm (*Bulusu et al., 2014*) developed as part of the canSAR drug discovery resource (*Tym et al., 2016*). In short, the algorithm identifies up to 10 cavities on a 3D-structure and measures ~30 geometric and physicochemical properties for each of these cavities. Such properties include the volume, enclosure, depth, and complexity of the cavity, as well as the number of hydrogen-bond donors and acceptors, polarity distribution and the projected ligand-binding energy of the cavity.

## Acknowledgements

We thank Anna C Groner, Jin Zhao, Jutta Stober and Rebecca Seeger for technical help and fruitful advice and assistance. A big thanks to Noriko Uetani for graphical support. We also thank the Molecular Biology Core Facility (Dana-Farber Cancer Institute), particularly Zach Herbert, for help with Illumina sequencing. This research was support by the Claudia Adams Barr Foundation (LC), the Prostate Cancer Foundation (LC, AS, JB, ACBC, MB) and the Deutsche Krebshilfe (ACBC). AS holds a joint Prostate Cancer UK and Academy of Medical Sciences Award and is supported by the Medical Research Council. LY was supported by a research grant from the KIT BioInterfaces International Graduate School (BIF-IGS) to ACBC and GUN. EAN is supported by a joined scholarship from the Government of Ghana and the DAAD (Germany). GUN, SB and US are supported by the German Science Foundation (DFG) (GRK 2039). XS is supported by MINECO and the European Research Council (contract number 648201). MB is a consultant for GTx and Novartis. He receives sponsored research support from Novartis.

## Additional information

### Funding

| Funder | Author |
| --- | --- |
| Prostate Cancer Foundation | Laura Cato<br>Adam Sharp<br>Johann S de Bono<br>Andrew CB Cato<br>Myles Brown |
| Deutsche Krebshilfe | Andrew CB Cato |
| Barr Foundation | Laura Cato |
| Prostate Cancer UK | Adam Sharp |
| Medical Research Council | Adam Sharp |

The funders had no role in study design, data collection and interpretation, or the decision to submit the work for publication.

### Author contributions

Laura Cato, Conceptualization, Data curation, Formal analysis, Funding acquisition, Validation, Investigation, Visualization, Methodology, Writing—original draft, Writing—review and editing; Antje Neeb, Data curation, Formal analysis, Validation, Investigation, Methodology, Writing—review and editing; Adam Sharp, Resources, Data curation, Formal analysis, Funding acquisition, Validation, Investigation, Visualization, Methodology, Writing—review and editing; Victor Buzón, Resources, Investigation, Visualization, Methodology; Scott B Ficarro, Claudia Muhle-Goll, Resources, Software, Investigation, Visualization, Methodology; Linxiao Yang, Resources, Data curation, Software, Formal analysis, Investigation, Visualization, Methodology; Nane C Kuznik, Emmanuel A Ntim, Jaice T Rottenberg, Liubov Shatkina, Claudia Ester, Investigation; Ruth Riisnaes, Daniel Nava Rodrigues,

Resources, Data curation, Investigation; Olivier Armant, Pasquale Rescigno, Resources, Software, Investigation; Victor Gourain, Software, Investigation; Guillaume Adelmant, Resources, Software, Investigation, Methodology; Thomas Westerling, Data curation, Methodology; David Dolling, Software; Ines Figueiredo, Resources, Investigation; Friedrich Fauser, Investigation, Methodology; Jennifer Wu, Resources, Software; Burkhard Luy, Holger Puchta, Uwe Strähle, Resources, Supervision; Jakob Troppmair, Nicole Jung, Stefan Bräse, Resources; Jarrod A Marto, Bissan Al-Lazikani, Resources, Software, Supervision; Gerd Ulrich Nienhaus, Xavier Salvatella, Resources, Software, Supervision, Writing—review and editing; Johann S de Bono, Resources, Software, Supervision, Funding acquisition, Writing—review and editing; Andrew CB Cato, Myles Brown, Conceptualization, Resources, Supervision, Funding acquisition, Writing—original draft, Writing—review and editing

### Author ORCIDs
Laura Cato (iD) http://orcid.org/0000-0002-7072-4368
Gerd Ulrich Nienhaus (iD) https://orcid.org/0000-0002-5027-3192
Xavier Salvatella (iD) http://orcid.org/0000-0002-8371-4185
Andrew CB Cato (iD) http://orcid.org/0000-0001-8508-3834
Myles Brown (iD) http://orcid.org/0000-0002-8213-1658

### Decision letter and Author response
Decision letter https://doi.org/10.7554/eLife.27159.040
Author response https://doi.org/10.7554/eLife.27159.041

## Additional files

### Supplementary files
• Transparent reporting form
DOI: https://doi.org/10.7554/eLife.27159.036

### Major datasets
The following dataset was generated:

| Author(s) | Year | Dataset title | Dataset URL | Database, license, and accessibility information |
|---|---|---|---|---|
| Cato L, Neeb A, Armant O, Gourain V, Cato AC, Brown M | 2017 | Targeting the androgen receptor N-terminus via the cochaperone Bag-1L | https://www.ncbi.nlm.nih.gov/geo/query/acc.cgi?acc=GSE89939 also see https://www.ncbi.nlm.nih.gov/geo/query/acc.cgi?acc=GSE89917 | Publicly available at the NCBI Gene Expression Omnibus (accession no: GSE89939) |

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
