## [Decision Letter]

Thank you for submitting your article "Transactivation by the intrinsically disordered androgen receptor N-terminal domain is enabled by the cochaperone Bag-1L" for consideration by *eLife*. Your article has been reviewed by three peer reviewers, one of whom, Chi Van Dang, is a member of our Board of Reviewing Editors, and the evaluation has been overseen by Charles Sawyers as the Senior Editor. The following individual involved in review of your submission has agreed to reveal their identity: Steven Metallo (Reviewer #2).

The reviewers have discussed the reviews with one another and the Reviewing Editor has drafted this decision to help you prepare a response to the critiques, which are quite significant.

Summary:

The manuscript by Cato et al. provides an analysis of the interaction of the androgen receptor (AR) transactivation domain (AF-1) with the chaperone BAG1L, which is demonstrated to be required for AR's full activity. As such, BAG1L may play a role in castrate-resistant prostate cancer (CRPC). The authors document that BAG1L is necessary for proper conformation of AR through FRET reporters and for DNA binding of AR to target sites. Using a peptide screen and mutational analysis a BAG1L domain required for docking with AR was identified. Further structural analysis and reconstitution of BAG1L KO cells with WT or mutant BAG1L demonstrated the necessity of this interaction domain of BAG1L or AR function in cells. Structural analysis further uncover a potentially druggable BAG1L pocket flanking this interaction domain. Key findings are supported by data to demonstrate: that loss of Bag-1L inhibits AR activity (as monitored by transcriptome profiling and ChIP-seq), prostate cancer growth using hormone-dependent model systems; that the Bag-1L/AR interaction is mediated by discrete domains in Bag-1L within a potentially targetable Bag domain; that Bag-1L enhances prostate cancer growth; and that in clinical specimens, nuclear Bag-1L levels increase during disease progression, predicting a reduced clinical benefit from the ligand-targeting agent abiraterone.

Essential revisions:

1) The concept that targeting Bag-1L to suppress AR activity and prostate cancer growth is novel, but additional experimentation that would enhance confidence in these findings would be to assess the Bag-1L requirement in additional model systems. The present study largely utilized a single model of hormone-therapy sensitive disease (LNCaP). Demonstration that the Bag-1L is maintained in other models, especially in castrate-resistant disease, should be included.

2) The model in Figure 6 does not seem to be dependent on new, specific data from the manuscript. The title of the current manuscript "Transactivation by the intrinsically disordered androgen receptor N-terminal domain is enabled by the cochaperone Bag-1L" could be anticipated from the authors' previous papers especially Shatkina et al., 2003 "The Cochaperone Bag-1L Enhances Androgen Receptor Action via Interaction with the NH2-Terminal Region of the Receptor". The authors are fully transparent about these previous results, stating clearly (e.g. in subsection “The BAG domain of Bag-1L binds the tau-5 region of AR”) that they have already shown that the BAG domain interacts primarily with AR tau-5. While the individual experiments each adds some information, they do not, together, lead to a novel conclusion.

3) The authors take advantage of previous AR-NTD assignments to demonstrate clear loss of signal in tau-5 of AR (Figure 3) that is specifically dependent on Bag-1L – indicative of interaction in this region. The authors have previously shown that the tau-5 domain has substantial helical propensity. Here the authors indicate that loss of signal in the tau-5 domain is "suggesting that wild-type BAG domain interacts with a specific conformation of this sub-domain." (subsection “The Bag-1L:AR interaction alters the structure of the AR NTD and is druggable”) There are no changes in the chemical shifts of the remaining, observable Bag-1L, indicating that the observed conformational ensemble is the same. There is no direct indication from the experiments that Bag-1L interacts with a specific conformation of tau-5 (either through conformational selection or induced fit). Binding of IDR regions often involve coupled folding and binding but there are examples of IDRs engaged in specific binding without folding. From the BAG domain side, there are NMR spectra (Figure 2—figure supplement 1) but no indication of experiments looking at potential chemical shift changes in the presence of AR. Those experiments may have indicated specific interacting regions on the BAG domain. The NMR experiments presented do not provide substantially new information. This manuscript would be more compelling if initial tool compounds were identified from the conceptual framework of inhibiting a chaperone to keep an intrinsically disordered protein inactive.

4) The identification of the triple mutant (CMut) has experimental utility, however it does not provide information about the actual BAG-AR interaction. As the substantial changes in the NMR make clear (Figure 2—figure supplement 1) and the authors point out (in subsection “The BAG domain of Bag-1L binds the tau-5 region of AR”) there is an overall change in the BAG domain caused by these mutations so that their actual positions are not necessarily important. The canSAR screen for binding sites (Figure 3) identified a potential "ligandable" site (subsection “The Bag-1L:AR interaction alters the structure of the AR NTD and is druggable”), although no defined "druggable" sites. This screen was not dependent on the present data. From the data it is not possible to tell if the presence of the Lys mutants of CMut at or near the edges of this cavity is coincidental or meaningful. Mutations within the cavity may have provided clearer evidence or whether the site is potentially useful.

5) Bag-1L has multiple functions outside AR regulation that may contribute to the anti-tumor effects observed in the xenograft studies. Investigation of the impact of Bag-1L alteration in the xenograft tumors should be explored (eg impact on AR signaling, proliferation, apoptosis, etc). Given the role of Bag-1L as a cochaperone, and nucleotide exchange factor for Hsp70, it would be important to determine the role of Hsp70 in the observed effects.

---

## [Author Response]

Essential revisions:1) The concept that targeting Bag-1L to suppress AR activity and prostate cancer growth is novel, but additional experimentation that would enhance confidence in these findings would be to assess the Bag-1L requirement in additional model systems. The present study largely utilized a single model of hormone-therapy sensitive disease (LNCaP). Demonstration that the Bag-1L is maintained in other models, especially in castrate-resistant disease, should be included.

We have investigated the function of Bag-1L in LNCaP95 and LNCaP-abl cells, two castrate-resistant prostate cancer (CRPC) models derived from the hormone-sensitive LNCaP cell line. Overexpression of wild-type Bag-1L did not affect CRPC cell line growth, presumably due to endogenously elevated levels of Bag-1L in these lines compared to the parental LNCaP cells. However, we observed a dominant negative effect on cell growth when overexpressing the Bag-1L CMut, which is defective in AR interaction. This result is in agreement with findings in LNCaP cells and suggests that Bag-1L remains important for AR function at CRPC status. The described data has been included in Figure 5 and Figure 5—figure supplement 3 to 5. The corresponding text (subsection “Bag-1L promotes androgen-dependent and -independent PCa growth”, figure legends for Figure 5 and Figure 5—figure supplement 3) and Methods and material section (subsection “Cell line preparation and maintenance”) have been updated to represent the new results.

2) The model in Figure 6 does not seem to be dependent on new, specific data from the manuscript. The title of the current manuscript "Transactivation by the intrinsically disordered androgen receptor N-terminal domain is enabled by the cochaperone Bag-1L" could be anticipated from the authors' previous papers especially Shatkina et al., 2003 "The Cochaperone Bag-1L Enhances Androgen Receptor Action via Interaction with the NH2-Terminal Region of the Receptor". The authors are fully transparent about these previous results, stating clearly (e.g. in subsection “The BAG domain of Bag-1L binds the tau-5 region of AR”) that they have already shown that the BAG domain interacts primarily with AR tau-5. While the individual experiments each adds some information, they do not, together, lead to a novel conclusion.

The novel findings in this study are (1) the discovery of specific residues in the BAG domain of BAG-1L required for the interaction with the AR tau-5 domain; (2) the impact of Bag-1L on the AR cistrome; (3) demonstration that the effect of Bag-1L on AR function is through a change in AR conformation; (4) comprehensive analysis of the Bag-1L interactome and its alterations in response to BAG domain mutation; (5) identification of a potential druggable pocket in Bag-1L. In addition, in the revised manuscript we have included data utilizing a tool Bag-1 inhibitor that prevents the Bag-1L:AR interaction. We agree with the reviewers that the present title does not highlight the novelty of the present manuscript. To address this concern, we have changed the title of the current manuscript to: “Development of Bag-1L as a Therapeutic Target in Androgen Receptor-Dependent Prostate Cancer”. Moreover, we have updated the model (Figure 7) and have made minor changes to the text and figure legend relevant to Figure 7.

3) The authors take advantage of previous AR-NTD assignments to demonstrate clear loss of signal in tau-5 of AR (Figure 3) that is specifically dependent on Bag-1L – indicative of interaction in this region. The authors have previously shown that the tau-5 domain has substantial helical propensity. Here the authors indicate that loss of signal in the tau-5 domain is "suggesting that wild-type BAG domain interacts with a specific conformation of this sub-domain." (subsection “The Bag-1L:AR interaction alters the structure of the AR NTD and is druggable”) There are no changes in the chemical shifts of the remaining, observable Bag-1L, indicating that the observed conformational ensemble is the same. There is no direct indication from the experiments that Bag-1L interacts with a specific conformation of tau-5 (either through conformational selection or induced fit). Binding of IDR regions often involve coupled folding and binding but there are examples of IDRs engaged in specific binding without folding. From the BAG domain side, there are NMR spectra (Figure 2—figure supplement 1) but no indication of experiments looking at potential chemical shift changes in the presence of AR. Those experiments may have indicated specific interacting regions on the BAG domain. The NMR experiments presented do not provide substantially new information. This manuscript would be more compelling if initial tool compounds were identified from the conceptual framework of inhibiting a chaperone to keep an intrinsically disordered protein inactive.

We agree with the reviewers that we cannot confirm that the interaction between the Bag-domain and AR tau-5 occurs in a specific conformation. Although we observe a decrease of about 50% intensity in the resonances of AR (between residues 375 and 448) in the presence of Bag-1L, no chemical shifts changes were observed. Based on this data, we conclude that the AR tau-5 domain is the main interaction site, because the decreases in residue intensities that we observe in the rest of the AR-AF-1 domain are smaller. As previously shown by our characterization of the structural properties of AR-AF-1, the region of sequence that is involved in the interaction with Bag-1L has substantial secondary structure when unbound. We think that it is plausible that these structural properties play a role in the described interaction with Bag-1L, but since we did not measure the chemical shifts of the bound state of AR, we cannot substantiate this claim. For this reason, we agree with the reviewers that we should rephrase the sentence: “suggesting that wild-type BAG domain interacts with a specific conformation of this sub-domain”, which has now been updated in the manuscript (subsection “The Bag-1L:AR interaction alters the structure of the AR NTD and is druggable”). The proposed experiment of BAG domain NMR in response to the addition of AR tau-5 suggested by this reviewer may provide additional information on binding of these proteins, but these are technically very challenging studies due to the nature of the two proteins. We therefore propose to attempt these experiments in a separate study.

In response to the comments regarding the use of tool compounds, we have now added some novel data on the function of the BAG domain inhibitor Thio-2 (Figure 4). We have added an appropriate section in the text to explain those results (subsection “The Bag-1L:AR interaction can be inhibited by the thioflavin Thio-2”), as well as a relevant figure legend, a relevant section in the Discussion. We have also made some small additions to the Materials and methods.

4) The identification of the triple mutant (CMut) has experimental utility, however it does not provide information about the actual BAG-AR interaction. As the substantial changes in the NMR make clear (Figure 2—figure supplement 1) and the authors point out (in subsection “The BAG domain of Bag-1L binds the tau-5 region of AR”) there is an overall change in the BAG domain caused by these mutations so that their actual positions are not necessarily important. The canSAR screen for binding sites (Figure 3) identified a potential "ligandable" site (subsection “The Bag-1L:AR interaction alters the structure of the AR NTD and is druggable”), although no defined "druggable" sites. This screen was not dependent on the present data. From the data it is not possible to tell if the presence of the Lys mutants of CMut at or near the edges of this cavity is coincidental or meaningful. Mutations within the cavity may have provided clearer evidence or whether the site is potentially useful.

The wild-type and CMut BAG domains have indistinguishable secondary structure, as shown in ^13^C correlation nuclear magnetic resonance (NMR) spectra to compare Cα and Cβ shifts (Figure 2—figure supplement 5). This suggests that the extent of α-helix formation essential for the interaction with AR is essentially unchanged for the two proteins. However, more than one third of the residues that make up the three antiparallel, helix bundles of the wild-type BAG domain shifted to new positions or demonstrated reduced signal intensities in ^1^H^15^N-HSQC NMR spectra in response to the K231/232/279A mutations (Figure 2—figure supplement 6). As suggested in our manuscript, this is most likely due to a destabilization of the BAG domain of Bag-1L caused by the three mutations. We agree with the reviewers that the NMR analysis presented here cannot be utilize to confirm the direct interaction of Bag-1L (through three lysine residues within the BAG domain) and the AR tau-5, given the points raised above. We can however confirm that the BAG domain of Bag-1L is necessary for the AR interaction, as it can no longer bind to AR in its unfolded conformation, which is the result of the triple mutation (CMut). We concur that other mutations within the BAG domain (such as mutations situated directly in the druggable cavity) might have similar effects. To address this, we have added data, where we have tested additional point mutations (and combination of point mutations) for their efficacy in inhibiting the Bag-1L-mediated AR transactivation in functional studies. The results are presented in Figure 2—figure supplement 4 and the accompanying text can be found in subsection “The BAG domain of Bag-1L binds the tau-5 region of AR” and Figure 2—figure supplement 4 figure legend. None of these additional mutations could be mapped to the druggable pocket of the Bag-1L BAG domain, although they all impaired the Bag-1L-mediated activation of AR AF1 activity. Conversely, several other mutations of the BAG domain (particularly in helix 1 and 2, such as residue 279) that did map directly onto, or the edge of the druggable pocket (Figure 3), did not inhibit binding to AR tau-5 or repress AR transactivation (Figure 2). We therefore believe that the overlap between the K231/232/279A mutation and the druggable cavity is indeed meaningful and not purely coincidental.

5) Bag-1L has multiple functions outside AR regulation that may contribute to the anti-tumor effects observed in the xenograft studies. Investigation of the impact of Bag-1L alteration in the xenograft tumors should be explored (eg impact on AR signaling, proliferation, apoptosis, etc). Given the role of Bag-1L as a cochaperone, and nucleotide exchange factor for Hsp70, it would be important to determine the role of Hsp70 in the observed effects.

We agree with the reviewers that Bag-1L has multiple functions outside AR regulation that may contribute to its anti-tumor effects in the xenograft studies. The many functions of Bag-1L originate mainly from its BAG domain, a protein binding surface for many cellular factors and signaling molecules. To explore the impact of Bag-1L alteration in AR signaling, apoptosis or proliferation, as suggested by the reviewers, we performed extensive in vitro mutagenesis studies of the BAG domain. This allowed us to separate the different function of Bag-1L and to specifically focus on its (Hsp70-independent) role on AR action. In addition to extensively testing the effect of Bag-1L alterations (through the study of CMut compared to wild-type Bag-1L) on AR-mediated cell growth in vitro and in xenograft studies (Figure 5), we have additionally carried out ChIP-seq and RNA-seq in LNCaP cells expressing the wild-type or CMut Bag-1L protein. We have now added the relevant data in Figure 2—figure supplement 2 and 3 and have updated the text accordingly to reflect these additions (i.e. Results subsection “The BAG domain of Bag-1L binds the tau-5 region of AR” and figure legends). Given the reproducibility of the growth experiments in vitro and in xenograft studies, we believe that the above-described functional experiments would result in similar outcomes and therefore would be redundant for inclusion in this manuscript. We have instead started experiments in which we have crossed Bag-1 knockout mice with two different prostate tumor models (TRAMP and PTEN knockout). Here where we see a significant decrease in tumor formation. We now plan to extend these studies to include mice with Bag-1L alterations. We believe that such animal studies will provide more detailed results that go beyond obtainable results from xenograft studies. However, such studies will take considerable time to complete and therefore cannot be included in this manuscript.

On the issue of Bag-1L functioning as a cochaperone of Hsp70, we have generated additional mutations, including a mutant that is defective in binding both AR and Hsp70 (D293A). Although such a mutation may implicate Hsp70 in the Bag-1L-mediated action on AR function, our main aim in this work was to identify mutations that would inhibit the BAG domain:AR interaction, independent of Hsp70. In line with recent findings of heat shock protein-independent functions of cochaperones (Taipale et al., 2014), we believe Bag-1L has a unique ability in regulating AR tau-5 activity, independent on Hsp70. In this connection, it is important to note that the residues of the Bag-1L CMut protein, which disrupt the BAG domain:AR tau-5 interaction, lie within helices 1 and 2 and are therefore distinct from the Hsp70 interaction sites in helix 3 (Sondermann et al., 2001).